# Unsupervised Multi-Agent Diversity With Wasserstein Distance

## Abstract

In cooperative Multi-Agent Reinforcement Learning (MARL), agents sharing policy network parameters are observed to learn similar behaviors, which impedes efficient exploration and easily results in the local optimum of cooperative policies. In order to encourage multi-agent diversity, many recent efforts have contributed to distinguishing different trajectories by maximizing the mutual information objective, given agent identities. Despite their successes, these mutual information-based methods do not necessarily promote exploration. To encourage multi-agent diversity and sufficient exploration, we propose a novel Wasserstein Multi-Agent Diversity (WMAD) exploration method that maximizes the Wasserstein distance between the trajectory distributions of different agents in a latent representation space. Since the Wasserstein distance is defined over two distributions, we further extend it to learn diverse policies for multiple agents. We empirically evaluate our method in various challenging multi-agent tasks and demonstrate its superior performance and sufficient exploration compared to existing state-of-the-art methods.

## 1 Introduction

Multi-Agent Reinforcement Learning (MARL) has shown promise in addressing various multi-agent challenges, such as multiplayer video games (Vinyals et al., 2019) and autonomous cars (Cao et al., 2012), attracting growing interest in recent years. MARL facilitates efficient collaboration by training multiple agents together towards maximizing team rewards. Yet, there are still many challenges such as partial observation constraints and high scalability requirements, when learning effective cooperative policies for agents in complex multi-agent tasks. To resolve these issues, recent works commonly employ the Centralized Training with Decentralized Execution (CTDE) framework (Lowe et al., 2017) where agents make decisions based on local observations using a decentralized policy jointly trained with global information, ensuring robust and stable performance.

The CTDE framework develops distinct decentralized policies for each agent, but training numerous policy network parameters can be inefficient. Thus, parameter sharing has become universal, allowing agents to share the same policy network parameters for action decision-making. This practice significantly reduces the number of parameters, leading to lower computational cost and speeding up training. Additionally, parameter sharing promotes experience sharing during centralized training, fostering robust policy learning and improving overall efficiency (Wang et al., 2020b).

Given these benefits, various MARL algorithms integrate parameter sharing, including value-decomposition approaches (Iqbal et al., 2021; Yang et al., 2021; Wang et al., 2020a; Sunehag et al., 2018; Rashid et al., 2018) and policy gradients (Ma et al., 2021; Wang et al., 2020d; Ndousse et al., 2021; Zhang et al., 2021). However, shared policy parameters can lead to homogeneous behaviors among agents, hindering multi-agent diversity and efficient exploration (Hu et al., 2022). In challenging multi-agent tasks, extensive exploration and diverse policies are crucial. For example, in a football game, agents must adopt varied roles and strategies for effective collaboration and goal scoring.

To address this issue, previous methods aim to promote identity-aware multi-agent diversity by maximizing mutual information between trajectories and agent identities (Jiang and Lu, 2021; Li et al., 2021; Charakorn et al., 2023; Jo et al., 2024). While these methods do learn trajectories that are mutually different, the mutual information objective cannot measure how different the

learned trajectories are. Slight differences between trajectories are enough to maximize the mutual information objective, which does not necessarily encourage exploration.

To encourage multi-agent diversity and sufficient exploration, we proposes a novel Wasserstein Multi-Agent Diversity (WMAD) exploration method. Our method relies on the Wasserstein distance (Villani et al., 2009), a metric-aware quantity to measure the distance between two different distributions. Wasserstein distance has drawn increasing attention in unsupervised reinforcement learning to encourage agents to sufficiently explore the state space, resulting in learning a diverse set of skills (Park et al., 2024). The motivation behind our method is that as the Wasserstein distance naturally quantifies the differences between different distributions, we can enlarge the distance between the trajectory distributions of different agents by maximizing the Wasserstein distance. Therefore, compared to mutual information-based methods, our method can lead to more diverse policies and sufficient exploration.

Our contributions can be summarized as follows: First, because of the similar trajectories generated by agents sharing the same policy network parameters, the Wasserstein distance, which measures the distance between different agents' trajectories, tends to approach zero. This implies that the Wasserstein distance cannot provide effective feedback for policy learning. To solve this issue, we consider a latent representation space in order to make the Wasserstein distance meaningful. To construct the representation space, we propose a next-step prediction method based on Contrastive Predictive Coding (CPC) (Oord et al., 2018) to learn distinguishable trajectory representations. Second, due to the high computation cost of calculating the Wasserstein distance, we propose a novel Gaussian kernel method to optimize dual functions of the Wasserstein distance, significantly reducing the computational cost. Third, we extend the Wasserstein distance to multiple policy learning by introducing a nearest neighbor intrinsic reward. We further integrate our method with QMIX. Fourth, we show the outperformance of our method against existing state-of-the-art methods by testing it in various challenging multi-agent tasks.

## 2 BACKGROUNDS

### 2.1 MULTI-AGENT SYSTEM

We consider modeling the fully cooperative multi-agent Decentralized Partially Observable Markov Decision Process (Dec-POMDP) (Oliehoek and Amato, 2015), defined as a tuple $\langle A, S, U, P, R, O, \Omega, \gamma \rangle$. Here, $A$ denotes a set of $|A|$ agents, $s \in S$ represents the global state of the environment, and $U$ stands for the set of agents' actions. At each time step, each agent $a$ receives an observation $o^a \in \Omega$ drawn from the function $O(s, a)$ and subsequently selects an action $u^a \in U$. All agents' actions collectively form a joint action $\boldsymbol{u}$, leading the environment to transition to the next state $s'$ based on the probability drawn from the transition function $P(s' \mid s, \boldsymbol{u})$. Simultaneously, the environment provides the agents with a shared team reward $r = R(s, \boldsymbol{u})$. $\gamma \in [0, 1)$ is the reward discount factor. The observation-action pairs $\langle o^a, u^a \rangle$ of agent $a$ during an episode constitute its trajectory $\tau^a \in \mathcal{T}$. Each agent $a$ learns its individual policy $\pi^a(u^a \mid \tau^a)$, contributing to the formation of a joint policy $\boldsymbol{\pi}$, aimed at maximizing the joint action-value function $Q^{\boldsymbol{\pi}}(s, \boldsymbol{u}) = \mathbb{E}_{s_{0:\infty}, \boldsymbol{u}_{0:\infty}} [\sum_{t=0}^{\infty} \gamma^t r_t \mid s_0 = s, \boldsymbol{u}_0 = \boldsymbol{u}, \boldsymbol{\pi}]$.

### 2.2 WASSERSTEIN DISTANCE

The Wasserstein distance formulates an optimal transport problem that measures the distance or discrepancy between two probability distributions (Villani et al., 2009). Given two probability distributions $p$ and $q$ over domains $\mathcal{X} \subseteq \mathbb{R}^m$ and $\mathcal{Y} \subseteq \mathbb{R}^n$ respectively, the Wasserstein distance with a cost function $c(x, y)$: $\mathcal{X} \times \mathcal{Y} \to \mathbb{R}$ is defined as:

$$\mathcal{W}_c(p, q) = \inf_{\gamma \in \Gamma(p,q)} \int_{\mathcal{X} \times \mathcal{Y}} c(x, y) \mathrm{d}\gamma(x, y) \tag{1}$$

where $\Gamma(p, q)$ is a set of all possible couplings of distributions $p$ and $q$ over the product space $\mathcal{X} \times \mathcal{Y}$. The probability distributions $p$ and $q$ are the marginals of the coupling $\gamma(x, y)$ over space $\mathcal{X}$ and $\mathcal{Y}$, respectively, i.e., $\int_{\mathcal{M}} \gamma(x, y) \mathrm{d}y = p(x)$ and $\int_{\mathcal{M}} \gamma(x, y) \mathrm{d}x = q(y)$.

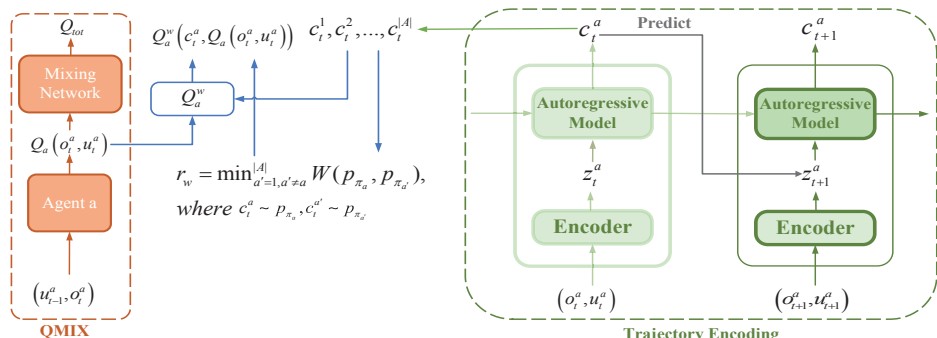

Figure 1: Architecture of WMAD.

In practice, we adopt a smoothed Wasserstein distance $\tilde{W}_c(p, q)$, which is a variant of the Wasserstein distance that can help mitigate the effects of outliers or noise in the distributions and lead to more stable optimization results (Genevay et al., 2016). It is intractable to compute the $\tilde{W}_c(p, q)$ directly, we resort to a traceable smoothed Fenchel-Rockafellar duality (Villani et al., 2009),

$$\tilde{W}_c(p, q) = \sup_{\mu, \nu} \mathbb{E}_{x \sim p(x), y \sim q(y)} \left[ \mu(x) - \nu(y) - \beta \exp \left( \frac{\mu(x) - \nu(y) - c(x, y)}{\beta} \right) \right] \quad (2)$$

where $\mu : \mathcal{X} \to \mathbb{R}$ and $\nu : \mathcal{Y} \to \mathbb{R}$ are dual functions on continuous domains. $\beta$ is a smoothing parameter. The dual form of the Wasserstein distance allows for the parametrization of dual functions, thereby mitigating the computational complexity of optimizing the optimal transport problem.

## 3 LIMITATIONS OF MI-BASED MULTI-AGENT DIVERSITY

To encourage multi-agent diversity, the most common approach adopted in prior work is to maximize the mutual information between trajectories $\tau$ and agent identities $i$ (Jiang and Lu, 2021; Li et al., 2021; Charakorn et al., 2023; Jo et al., 2024), which associates different trajectories with different agent identities. Agents can learn trajectories that are mutually distinct through maximizing the mutual information objective. The mutual information objective is based on the KL divergence, computed by a variational lower bound,

$$I(\tau; i) = D_{KL}(p(\tau, i) \| p(\tau)p(i)) \geq \mathbb{E}_{i, \tau} \left[ \log q_\theta(i \mid \tau) \right] - \mathbb{E}_i[\log p(i)], \quad (3)$$

where the distribution of agent identities $p(i)$ is a constant since the agent identity $i$ follows a uniform distribution. Thus, the objective of maximizing the mutual information can be achieved by maximizing the trajectory discriminator $q_\theta(i \mid \tau)$ parameterized by $\theta$, i.e., once agent trajectories can be successfully discriminated given agent identities, the maximum of the mutual information can be achieved. However, this category of methods share a limitation that the maximum of the mutual information can be easily obtained when the trajectories learned by agents are slightly different, which does not necessarily encourage the visitations of trajectories with large variations, resulting in insufficient exploration. This occurs because the KL divergence remains entirely agnostic to the metric of the underlying data distribution and unaffected by any invertible transformation (Ozair et al., 2019). The KL divergence is very sensitive to small changes in the data samples, which means that any slight difference is sufficient to maximize the KL divergence.

To address this issue, our method encourages multi-agent diversity by enlarging the Wasserstein distance between trajectory distributions of different agents in a latent representation space. Different from the KL divergence, Wasserstein distance explicitly measures the distance between different distributions. Thus, our method can drive agents to visit different trajectories as far as possible, leading to sufficient exploration. We refer the reader to Appendix D for a quantitative comparison between the Wasserstein distance and the KL divergence.

## 4 WASSERSTEIN MULTI-AGENT DIVERSITY

In this section, we detail our proposed Wasserstein Multi-Agent Diversity (WMAD). First, we present how to learn meaningful representations to generate effective feedback for the Wasserstein distance. Then, we show how to maximize the Wasserstein distance between different trajectory distributions in the latent representation space.

### 4.1 CONTRASTIVE PREDICTIVE TRAJECTORY REPRESENTATIONS

Due to the similar trajectories induced by the agents sharing the same policy network parameters, the Wasserstein distance between any two agents' trajectory distributions approaches zero, i.e., $W(X,Y) \to 0$, where $X$ and $Y$ respectively represent the trajectory distributions of two agents. Since we want the Wasserstein distance to produce effective feedback for agents to learn diverse policies, we propose a next-step prediction method based on Contrastive Predictive Coding (CPC) (Oord et al., 2018) to learn distinguishable trajectory representations.

Initially, we encode the observation-action pairs $x_t^a = (o_t^a, u_t^a)$ with a non-linear encoder $g_{\theta_e}$ into a latent embedding $z_t^a = g_{\theta_e}(x_t^a)$. Then, we use an autoregressive model $g_{\theta_g}$ to summarize all the latent embeddings and output the trajectory representation $c_t^a = g_{\theta_g}(z_{\leq t}^a)$ at timestep $t$. We simply denote $g_\theta = \{g_{\theta_e}, g_{\theta_g}\}$ to represent the overall trajectory encoder. For simplicity, we adopt standard architectures such as MLPs for $g_{\theta_e}$ and GRUs for $g_{\theta_g}$.

To train $g_\theta$ to learn distinguishable trajectory representations, we model a density ratio that preserves the underlying information between the trajectory representation $c_t^a$ and the next-step observation-action $x_{t+1}^a$:

$$f\left(x_{t+1}^a, c_t^a\right) \propto \frac{p\left(x_{t+1}^a \mid c_t^a\right)}{p\left(x_{t+1}^a\right)} \tag{4}$$

where $f\left(x_{t+1}^a, c_t^a\right) = \exp(g_{\theta_e}(x_{t+1}^a)^T W c_t^a) = \exp(z_{t+1}^{a\ T} W c_t^a)$ calculates the similarity between the next-step observation-action embedding $z_{t+1}^a$ and a linear transformation $W^T c_t^a$ with the parameter $W$ used for the next-step perdiction. Compared to modeling $p\left(x_{t+1}^a \mid c_t^a\right)$ directly by a generative method that requires to reconstruct every detail in $x_{t+1}^a$, modeling the density ratio has lower computation cost and is more effective in extracting shared information between $x_{t+1}^a$ and $c_t^a$. Moreover, we infer the latent embedding $z_{t+1}^a$ instead of the raw $x_{t+1}^a$, which avoids modeling high-dimensional observation-action space. To let $f\left(x_{t+1}^a, c_t^a\right)$ be proportional to the density ratio, inspired by CPC, given a set of next-step observation-action pairs of all agents $\mathcal{C} = \{x_{t+1}^{a'} = (o_{t+1}^{a'}, u_{t+1}^{a'})\}_{a'=1}^{|A|}$, we minimize a InfoNCE loss (Oord et al., 2018):

$$\mathcal{L}_N = -\mathop{\mathbb{E}}_{(c_t^a, \mathcal{C}) \sim \mathcal{D}}\left[\log \frac{f\left(x_{t+1}^a, c_t^a\right)}{\sum_{x_{t+1}^{a'} \in \mathcal{C}} f\left(x_{t+1}^{a'}, c_t^a\right)}\right] \tag{5}$$

By using the next-step observation-action pairs of other agents as noisy samples in Equation 5 and contrasting the trajectory representation $c_t^a$ with these noises, the trajectory representation $c_t^a$ stays close to its associated next-step observation-action embedding while being far away from other noisy embeddings. As a result, the trajectory encoder $g_\theta$ is trained by minimizing the InfoNCE loss to learn distinguishable trajectory representations.

### 4.2 WASSERSTEIN DISTANCE BETWEEN TRAJECTORY REPRESENTATIONS

We then encourage the exploration of diverse trajectories by maximizing the Wasserstein distance between the trajectory distributions of different agents in a latent representation space. Let $p_{\pi_1}$ and $p_{\pi_2}$ be the trajectory representation distributions of agent 1 and agent 2, respectively. The Wasserstein distance between $p_{\pi_1}$ and $p_{\pi_2}$ is defined as follows:

$$\tilde{W}_c(p_{\pi_1}, p_{\pi_2}) = \sup_{\mu, \nu} \mathbb{E}_{c_t^1 \sim p_{\pi_1}, c_t^2 \sim p_{\pi_2}}\left[\mu(c_t^1) - \nu(c_t^2) - \beta \exp\left(\frac{\mu(c_t^1) - \nu(c_t^2) - c(c_t^1, c_t^2)}{\beta}\right)\right] \tag{6}$$

where the cost function $c(c_t^1, c_t^2)$ is represented by the Euclidean distance between the points $c_t^1$ and $c_t^2$, i.e., $c(c_t^1, c_t^2) = \|c_t^1 - c_t^2\|$. It is notable that to compute the Wasserstein distance, we may simply parameterize dual functions with neural networks like previous works (Pacchiano et al., 2020; Dadashi et al., 2021; He et al., 2022; Park et al., 2024). However, this may lead to high computational costs in our multi-agent settings, as we need to compute the Wasserstein distance for each pair of agents. To learn optimal dual functions $\mu$ and $\nu$ to compute the Wasserstein distance with low computational costs, we resort to the kernel method (Hearst et al., 1998) that has been widely used in machine learning. Specifically, we consider representing dual functions with linear combinations of Gaussian kernel functions approximated by the random feature map (Rahimi and Recht, 2007). For example, let the dual function $\mu$ has the following form: $\mu(\mathbf{x}) = (\lambda^\mu)^\top \phi(\mathbf{x})$. For $\mathbf{x} \in \mathbb{R}^d$, $\phi(\mathbf{x}) = \frac{1}{\sqrt{m}} \cos(\mathbf{G}\mathbf{x} + \mathbf{b})$ represents a $m$-dimensional random feature map, where $\mathbf{G} \in \mathbb{R}^{m \times d}$ is a Gaussian with entries sampled from a normal distribution $\mathcal{N}(0, 1)$ and $\mathbf{b} \in \mathbb{R}^m$ with entries sampled from a uniform distribution $U(0, 2\pi)$. This means that when we optimize the dual function $\mu$, we only need to learn the dual vector $\lambda^\mu \in \mathbb{R}^m$, which significantly reduces the computational cost compared with parameterizing dual functions with computationally intensive neural networks.

To learn optimal dual functions, we perform stochastic gradient descent (SGD) over the Wasserstein distance objective in Equation 6. Given dual functions $\mu$ and $\nu$ that are modeled by kernels $\kappa$ and $\ell$, respectively, and trajectory representaion samples $\{c_t^1, c_t^2\} \sim (p_{\pi_1}, p_{\pi_2})$, we apply the chain rule to Equation 6 and the gradients with respect to $\lambda^\mu$ and $\lambda^\nu$ are

$$
\nabla_{(\lambda^\mu, \lambda^\nu)} \tilde{W}_c(p_{\pi_1}, p_{\pi_2}) =
$$
$$
\mathbb{E}_{c_t^1 \sim p_{\pi_1}, c_t^2 \sim p_{\pi_2}} \left[ \left( 1 - \exp \left( \frac{(\lambda^\mu)^\top \phi_\kappa(c_t^1) - (\lambda^\nu)^\top \phi_\ell(c_t^2) - C(c_t^1, c_t^2)}{\beta} \right) \right) \left( \begin{array}{c} \phi_\kappa(c_t^1) \\ -\phi_\ell(c_t^2) \end{array} \right) \right].
\tag{7}
$$

We approximate the expectation by averaging the function values over a batch of trajectory representation samples from the replay buffer that is used to store agent experiences during training.

As we have computed the value of the Wasserstein distance, we can view the Wasserstein distance as an intrinsic reward $r_w = W(p_{\pi_1}, p_{\pi_2})$, which enables us to deploy our method in MARL algorithms to maximize the Wasserstein distance. When the number of agents $|A|$ is more than two, the trajectory of an arbitrary agent should keep distance with any other agent. In practice, we empirically find that employing an intrinsic reward $r_w = \min_{a'=1, a' \neq a}^{|A|} W(p_{\pi_a}, p_{\pi_{a'}})$ for each agent to keep the trajectory of the current agent $a$ to be away from its nearest neighbor trajectory in a latent representation space can lead to better performance. The pseudocode for our method can be found in Appendix E.

## 4.3 Practical Learning Algorithm

We next show how to integrate our method with QMIX (Rashid et al., 2018), a state-of-the-art MARL algorithm. QMIX learns optimal individual policies, that maximizes shared team rewards, for agents through optimizing the joint action-value function $Q^\pi$ approximated by $Q_{tot}$, an output of a mixing network that monotonically mixes the agent utilities (where the policies are derived) of all agents. In QMIX, in order to maximize the Wasserstein distance-based intrinsic rewards, we cannot simply add each agent's intrinsic rewards to the shared team reward. More detailed explanations can be found in Appendix C. To integrate our method with QMIX, we additionally introduce an intrinsic utility network $Q_a^w$, which takes as input the agent utility $Q_a(o^a, u^a)$ and the trajectory representation $c_t^a$. We update $Q_a^w$ towards maximizing the intrinsic rewards by minimizing the TD loss as follows

$$
\mathcal{L}_{TD}^w = \mathbb{E}_{(o_t^a, u_t^a, o_{t+1}^a) \sim \mathcal{D}} \left[ (Q_a^w (c_t^a, Q_a(o_t^a, u_t^a)) - y)^2 \right],
$$
$$
where \quad y = r_w + \gamma \bar{Q}_a^w (c_{t+1}^a, \bar{Q}_a (o_{t+1}^a, u_{t+1}^a))
\tag{8}
$$

where $\bar{Q}_a^w$ and $\bar{Q}_a$ are target networks employed to stabilize training and $\mathcal{D}$ is the replay buffer for storing trajectory samples. $\mathcal{L}_{TD}^w$ can be seen as a regularizer that introduces an auxiliary gradient to the agent utility network $Q_a$ in order to learn diverse trajectories. We can thus get the total loss function

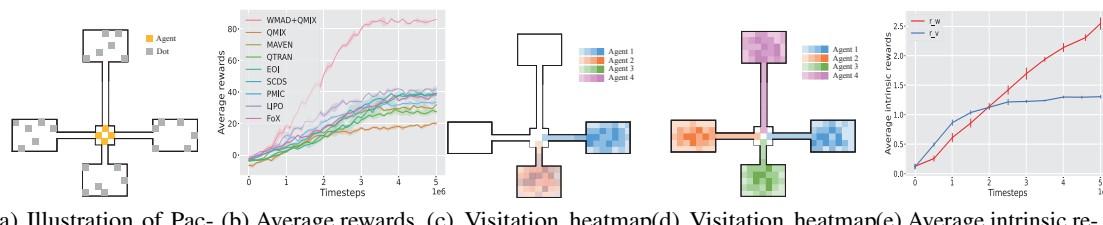

(a) Illustration of Pac-Men    (b) Average rewards    (c) Visitation heatmap of QMIX    (d) Visitation heatmap of WMAD    (e) Average intrinsic rewards

Figure 2: Performance comparison between our proposed WMAD and baselines in Pac-Men. We report both the mean and standard deviation of the performance tested across five random seeds.

$$\mathcal{L}_{total} = \mathcal{L}_{TD} + \alpha \mathcal{L}_{TD}^{w} \tag{9}$$

where $\mathcal{L}_{TD}$ is the TD loss of QMIX to train $Q_{tot}$ and $\alpha$ is a cofficient that changes the weight of $\mathcal{L}_{TD}^{w}$. As $\alpha \to 0$, our method converges to QMIX. Through minimizing $\mathcal{L}_{total}$, we train the overall framework of our method end-to-end in a centralized manner. As a result, agents learn their policies towards maximizing both team rewards and the Wasserstein distance between different agent's trajectory representation distributions. For policy gradient methods, we refer the reader to Appendix F where we integrate our method with the policy gradient-based method MAPPO.

## 5 EXPERIMENTS

In this section, we use challenging multi-agent tasks from Pac-Men, SMAC, and SMACv2 to demonstrate the outperformance of our method. We show comparison of our method against the state-of-the-art methods such as value-decomposition methods (QMIX (Rashid et al., 2018) and QTRAN (Son et al., 2019)) and mutual information-based exploration methods (MAVEN (Mahajan et al., 2019), EOI (Jiang and Lu, 2021), SCDS (Li et al., 2021), PMIC (Li et al., 2022), LIPO (Charakorn et al., 2023), and FoX (Jo et al., 2024)). Without loss of generality, the comparison results are shown with both the mean and standard deviation of the performance tested across five random seeds. For a fair comparison, we adopt the same common hyperparameters and policy network architecture across all methods. More training details and hyperparameters are provided in Appendix I.

### 5.1 PAC-MEN

We first test our method in Pac-Men, as illustrated in Figure 2a, to investigate the effectiveness of our method in encouraging multi-agent diversity. Pac-Men is a foraging game, where four agents initialized at the center of the maze try to eat the dots randomly distributed in four edge rooms. Agents can move to these rooms along paths of different lengths. Each agent only has a partial observation of 4×4 grid around them. The goal of the agent is to collect as many dots as possible to achieve more rewards. Notably, agents arriving at the same edge room may result in inefficient competition. They are expected to behave differently and move to different rooms.

The results shown in Figure 2b demonstrate the outperformance of our method compared to baselines. Through maximizing the Wasserstein distance between different trajectory distributions in a latent space, agents respectively move to the four edge rooms, as depicted by Figure 2d, leading to diverse policies and efficient cooperation. QMIX fails to learn diverse policies. As shown in Figure 2c, some agents adopt the same policy and move to the same edge room, resulting in poor performance. Some mutual information-based baselines such as EOI and SCDS employing the variational intrinsic rewards $r_v$ achieve similar performance. They may not find the edge room with the longest path due to inefficient exploration caused by the variational intrinsic rewards $r_v$, leading to sub-optimal performance. From Figure 2e, we note that the variational intrinsic reward $r_v$ converges quickly due to its metric-agnostic property, leading to insufficient incentives for exploration. Conversely, our Wasserstein distance-based metric-aware intrinsic reward $r_w$ can continuously provide effective reward signals for agents to encourage sufficient exploration.

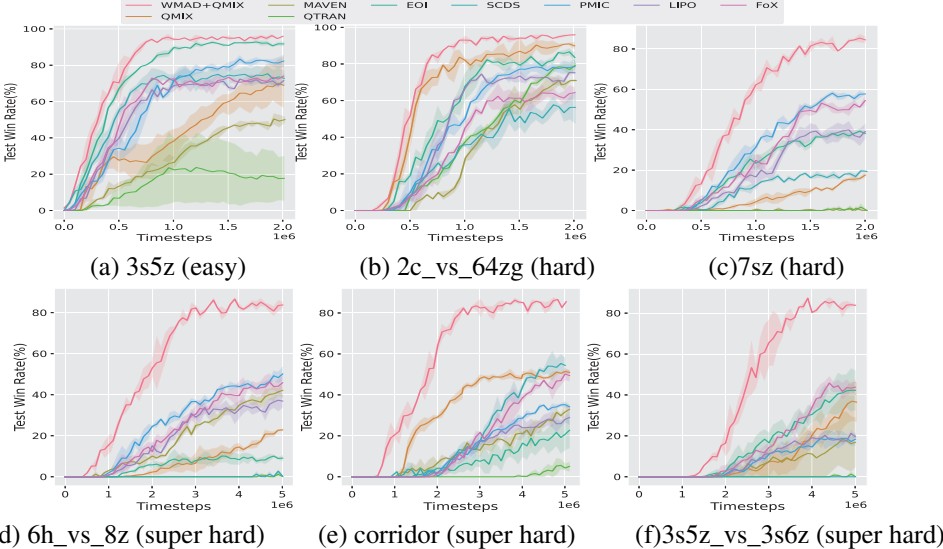

Figure 3: Performance comparison between our proposed WMAD and baselines in the SMAC scenarios.

## 5.2 SMAC

We then test our method on the StarCraft Multi-Agent Challenge (SMAC) (Samvelyan et al., 2019), a commonly used benchmark for evaluating cooperative MARL algorithms, consisting of various combat scenarios with different difficulties. We evaluate our method in 6 scenarios of SMAC including 3s5z (easy), 2c_vs_64zg (hard), 7sz (hard), 6h_vs_8z (super hard), corridor (super hard), and 3s5z_vs_3s6z (super hard). The version of SMAC adopted in our experiments is SC2.4.10. The performance comparison are not applicable across different SMAC versions.

As shown in Figure 3, our method maintains its outperformance in both easy and hard scenarios and significantly outperforms all baselines in the super hard scenarios. QMIX struggles to learn optimal cooperative policies in the super hard scenarios. However, our method can efficiently improve the performance of QMIX by encouraging multi-agent diversity. Compared to mutual information-based methods, our method achieves better performance due to the maximization of the metric-aware Wasserstein distance, leading to more sufficient exploration. We further present visualization examples of diverse policies learned by our method in the super hard scenarios in Appendix 7. The mutual information-based methods may not enable agents to learn trajectories with large variations. EOI does not result in satisfactory performance as the trajectory classifier employed in EOI overfits the agent identity information, impeding further exploration. Moreover, it is notable that our method also achieves satisfactory performance in the easy 3s5z scenario where agents sometimes need to behave in the same way to master the trick of 'focus fire', demonstrating that our method would not prevent the homogeneous behaviors that can lead to more environmental rewards. More experimental results related to such homogeneous behaviors can be found in Appendix G.2. These results reveal that our method efficiently balances exploration and exploitation, resulting in the learning of optimal cooperative policies.

**Stochasticity and Exploration** Although SMAC consists of many challenging scenarios, the agents may overfit the timesteps regardless of real environmental states Ellis et al. (2022) since the team compositions and the initial positions of units are the same in each episode. We further adopt the SMACv2 benchmark Ellis et al. (2022). SMACv2 introduces stochasticity by deploying random team compositions and random initial positions, which challenges agents to continuously explore optimal policies. The performance comparison of our method against baselines are shown in Figure 4. Our method achieves superior performance in all scenarios compared to the baselines. Our method significantly improves the performance of QMIX by introducing the Wasserstein distance objective as a regularizer to encourage multi-agent diversity. The mutual information-based methods do not yield satisfactory performance. We believe this is because the variational intrinsic reward adopted in

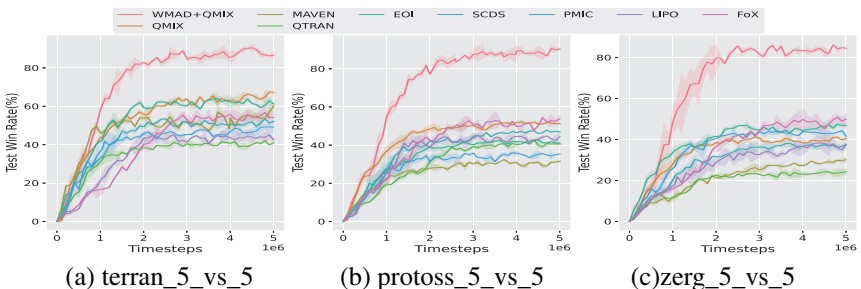

(a) terran_5_vs_5     (b) protoss_5_vs_5     (c)zerg_5_vs_5

Figure 4: Performance comparison between WMAD and baselines in the SMACv2 scenarios.

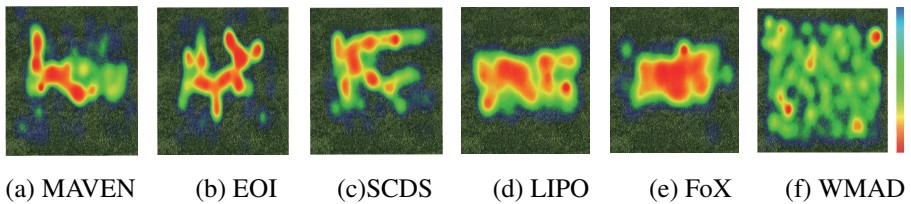

(a) MAVEN  (b) EOI  (c)SCDS  (d) LIPO  (e) FoX  (f) WMAD

Figure 5: Visitation heatmaps of different algorithms in the terran_5_vs_5 scenario.

these methods converge quickly when the trajectories of different agents are identified. As a result, it cannot provide effective feedback for agents to continuously explore. Instead, our method can continuously provide efficient Wasserstein distance-based intrinsic rewards to encourage exploration. This can be verified by the visitation heatmaps of agents trained by various methods shown in Figure 5. We observe that agents trained by our method achieve more extensive environmental exploration compared to those trained using baselines distributed only in partial areas.

## 5.3 ABLATION STUDY

We conduct ablation studies to evaluate the contributions of the main components in our method. To test the contribution of the autoregressive model employed to learn trajectory representations, we ablate the autoregressive model and only use the non-linear encoder $g_{\theta_e}$ regardless of the trajectory context. To measure the contribution of CPC, we design five variants: (i) employing a randomly initialized encoder with fixed parameters for encoding trajectories, (ii) learning trajectory representations by directly predicting the agent identities of various trajectories instead of employing the InfoNCE loss, (iii) learning trajectory representations by adopting a generative method to model $p\left(x_{t+1}^a \mid c_t^a\right)$ instead of modeling the density ratio, (iv) using CPC to predict the trajectory representation $c_{t+1}^a$ instead of the latent embedding $z_{t+1}^a$, and (v) adopting CPC to directly predict the raw observation-action $x_{t+1}^a$. To test the Wasserstein distance objective, we ablate the nearest neighbor intrinsic reward $r_w$ and use the Wasserstein distance between trajectory representation distributions of the current agent and another randomly selected agent and the average Wasserstein distance of all agents as intrinsic rewards, respectively.

We test these variants in the scenarios from SMAC, and the results are shown in Figure 6a. We note that the absence of any of the components employed in our method results in significant performance degradation. Encoding trajectory representations with a fixed encoder leads to poor performance, demonstrating the importance of using CPC to learn distinguishable trajectory representations. Moreover, learning trajectory representations by minimizing the identity prediction loss or learning a generative model is less efficient than our method. These methods do not necessarily learn distinguishable trajectory representations with large variations, thus the representations may not work properly in the Wasserstein distance objective to produce efficient feedback. Also, using the generative method leads to lower learning efficiency due to high computational cost. Using CPC to predict the trajectory representations or the raw observation-action does not lead to better performance than predicting the latent embeddings adopted in our method. The average Wasserstein distance does not yield satisfactory performance and even achieves worse performance than the random agent Wasserstein distance. As shown in Figure 6b, the average Wasserstein distance intrinsic

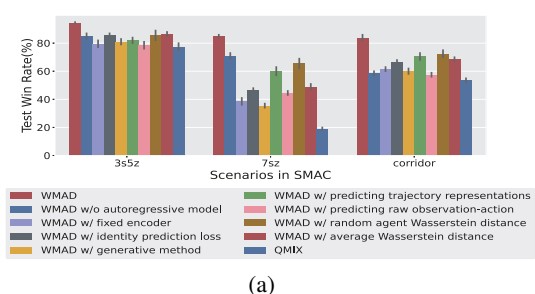 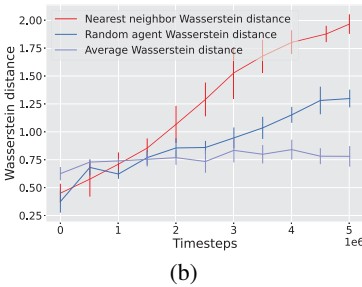

(a) (b)

Figure 6: (a) Performance comparison of different variants in the scenarios of SMAC. (b) Different kinds of Wasserstein distances.

rewards do not provide effective incentives to encourage multi-agent diversity. Instead, our nearest neighbor Wasserstein distance is more sensitive to the trajectory representation variations. Despite the performance degradation caused by different kinds of Wasserstein distances, these Wasserstein distance methods also lead to significant performance improvement over QMIX, demonstrating the robustness of our representation learning method. As the difficulty of the task increases, we note obvious performance degradation caused by the ablation of the autoregressive model, indicating that learning trajectory representations results in more robust performance, especially in hard tasks.

## 5.4 RELATED WORKS

Diversity within MARL aims to learn diversified policies among agents to encourage efficient exploration. To achieve this goal, numerous diversity-driven methods have proposed different intrinsic motivations or regularizers. RODE (Wang et al., 2020c) promotes diversity by assigning distinct actions to predefined roles; however, its effectiveness may decrease in scenarios with continuous actions and extensive action spaces. MAVEN (Mahajan et al., 2019) introduces a value-based approach that conditions agents' joint behaviors on a shared latent variable controlled by a hierarchical policy. EOI (Jiang and Lu, 2021) utilizes a supervised learning approach to promote agent individuality, employing a probabilistic classifier to predict agents' probability distributions based on their observations. SCDS (Li et al., 2021) concentrates on enhancing multi-agent diversity by optimizing mutual information between agent identities and trajectories. PMIC (Li et al., 2022) adopts a unique approach by maximizing the mutual information concerning superior cooperative behaviors while minimizing it regarding inferior behaviors. LIPO (Charakorn et al., 2023) uses policy compatibility as a proxy to learn diverse policies and diversifies agents' behaviors through the mutual information objective. FoX (Jo et al., 2024) proposes formation-based exploration, encouraging visitations of diverse formations by guiding agents to fully understand their current formations. Although these approaches show promise in enhancing multi-agent diversity, the KL divergence derived from the mutual information objective may lead to insufficient exploration. We refer the reader to Appendix A for related works about Wasserstein distance.

## 6 LIMITATIONS AND FUTURE DIRECTIONS

It is notable that the Wasserstein distance is determined by the cost function defining how the probability mass is transported. For simplicity, we choose the Euclidean distance as the cost function in all experimental environments. The cost function can be defined as different metrics across various tasks to measure the trajectory differences. However, choosing an appropriate cost function for the Wasserstein distance to solve specific multi-agent tasks can be challenging, which remains a goal for our future work.

## 7 CONCLUSION

In this paper, we propose a new WMAD exploration method. Unlike previous mutual information-based methods, our method maximizes the Wasserstein distance between the trajectory distributions

of different agents in a latent representation space learned by a next-step prediction method, leading to sufficient exploration. We deploy our method in MARL by introducing a nearest neighbor intrinsic reward based on the Wasserstein distance. The experimental results demonstrate that our method learns more diverse policies and leads to more sufficient exploration compared to mutual information-based methods. This simple yet effective method provides a novel idea of learning useful representations to promote exploration, which shows promising results in learning cooperative policies for challenging multi-agent tasks.

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

## A    RELATED WORKS ABOUT WASSERSTEIN DISTANCE

Wasserstein distance, emerging as an advanced measure of distribution dissimilarity, has garnered the attention of researchers from the machine learning community. Many generative models (Arjovsky et al., 2017; Ambrogioni et al., 2018; Patrini et al., 2020; Tolstikhin et al., 2018) have incorporated the Wasserstein distance objective and demonstrate the effectiveness of Wasserstein distance in scenarios where distributions become degenerate on a sub-manifold within pixel space. In reinforcement learning, the Wasserstein distance is used to evaluate the policy differences, supplanting commonly utilized KL divergence. BGPG (Pacchiano et al., 2020) uses the Wasserstein distance as a regularizer to improve the trust region policy optimization. PWIL (Dadashi et al., 2021) demonstrates the effectiveness of the Wasserstein distance in imitation learning by minimizing the Wasserstein distance between behavioral policies and expert policies. WURL (He et al., 2022) proposes using the Wasserstein distance to maximize the distance of state distributions to encourage the agent to sufficiently visit state space. METRA (Park et al., 2024) applies the Wasserstein distance to unsupervised pre-training to cover a compact latent space that is metrically associated with the state space. Our method is inspired by these methods and uses the metric-aware Wasserstein distance to encourage agents to learn more diverse policies in the domain of MARL.

## B    DIFFERENCES TO PERVIOUS MUTUAL INFORMATION-BASED METHODS

Prior work that maximizes the mutual information between trajectories and agent identities by maximizing the variational lower bound typically formulates a variational intrinsic reward:

$$r_v = \log q_\theta(i \mid \tau) - \log p(i), \tag{10}$$

The intrinsic reward $r_v$ intuitively encourages agents to visit different trajectories that can be successfully distinguished by the discriminator $q_\theta(i \mid \tau)$ given agent identities. However, the intrinsic reward $r_v$ cannot measure how different the trajectories are. To solve the issue, our method formulates a Wasserstein distance-based metric-aware intrinsic reward $r_w = \min_{a'=1, a' \neq a}^{|A|} W\left(p_{\pi_a}, p_{\pi_{a'}}\right)$ measuring the distance between the trajectory representation distributions of the current agent and its nearest neighbor. Therefore, through maximizing the intrinsic reward $r_w$, the Wasserstein distance can be enlarged, leading to more diverse trajectories.

## C    THE TD LOSS OF QMIX

The TD loss of QMIX to learn the optimal $Q_{tot}$ is defined as:

$$\mathcal{L}_{TD} = \sum_{i=1}^{b} \left[ \left( r + \gamma \max_{\mathbf{u}_{t+1}} \bar{Q}_{tot}\left(s_{t+1}, \mathbf{u}_{t+1}\right) - Q_{tot}(s_t, \mathbf{u}_t) \right)^2 \right] \tag{11}$$

where $\bar{Q}_{tot}$ is the target network and $b$ is the size of transition samples from the replay buffer $\mathcal{D}$. $r$ is the global reward shared among agents. Note that since all agent's policies are jointly trained by

minimizing the TD loss, we cannot simply apply each agent's intrinsic reward $r_w$ to the global reward $r$ to formulate a reward-shaping to independently train each agent's individual policy. That is why we need to learn an additional intrinsic utility network to maximize the intrinsic reward $r_w$.

## D  QUANTITATIVE COMPARISON BETWEEN THE WASSERSTEIN DISTANCE AND THE KL DIVERGENCE

To illustrate the difference between the Wasserstein distance and the KL divergence, we take a Gaussian distribution example. Let $p \sim \mathcal{N}\left(\mu_p, \sigma^2\right)$ and $q \sim \mathcal{N}\left(\mu_q, \sigma^2\right)$. As $\sigma \to 0$, the probability mass of $p$ and $q$ converges to their means, thus we can achieve the KL divergence between two distributions $p$ and $q$ $\lim_{\sigma \to 0} D_{\mathrm{KL}}(p\|q) = \infty$, which is independent of the specific means $\mu_p$ and $\mu_q$. The Wasserstein distance between $p$ and $q$ is $\lim_{\sigma \to 0} W(p, q) = |\mu_p - \mu_q|$. We note that the Wasserstein distance provides an explicit measurement of distance, whereas the KL divergence focuses only on distinguishability and has no relevance to the metric of the underlying data distribution. As a result, due to the metric-aware property of the Wasserstein distance, our method can not only encourage the visitations of different trajectories, as in the KL divergence, but also maximize the distance between diverse trajectories that leads to better trajectory space coverage and more sufficient exploration.

## E  PSEUDOCODE FOR WMAD

The pseudocode for WMAD is given in Algorithm 1.

---

**Algorithm 1:** Wasserstein Multi-Agent Diversity (WMAD)

---

Initialize dual functions $\mu$ and $\nu$. Initialize the joint policy $\boldsymbol{\pi} = \{\pi_a\}_{a=1}^{|A|}$.
Randomly initialize $Q_{tot}$ for QMIX.
**repeat**
  **for** *each episode* **do**
    Collect the trajectories of all agents $\boldsymbol{\tau}$ induced by the joint policy $\boldsymbol{\pi}$.
    Store them into a replay buffer $D$.
  **end for**
  Sample a batch of trajectories $\boldsymbol{\tau}$ from the replay buffer $D$.
  Train the trajectory encoder $g_\theta$ to learn trajectory representations by minimizing the InfoNCE loss given by Equation 5.
  Train dual functions $\mu$ and $\nu$ by SGD with the gradient given by Equation 7.
  Compute the intrinsic reward $r_w = \min_{a'=1, a' \neq a}^{|A|} W\left(p_{\pi_a}, p_{\pi_{a'}}\right)$ for each agent.
  Jointly train the policy $\pi_a$ for each agent by minimizing $\mathcal{L}_{total} = \mathcal{L}_{TD} + \alpha \mathcal{L}_{TD}^w$.
**until** $Q_{tot}$ is *converged*

---

## F  INTEGRATING WMAD WITH THE POLICY-BASED METHOD

We have implemented our method with the value-based method QMIX. Here, we illustrate the integration of our proposed WMAD with policy-based methods. Specifically, we integrate WMAD with MAPPO, a state-of-the-art policy-based MARL algorithm measured by SMAC. In MAPPO, all agents share an actor network and a critic network. As each agent learns its own critic, we can straightforwardly incorporate a shaped reward $r_{env} + \alpha r_w$ (where $r_{env}$ represents the environmental reward and $r_w$ denotes the Wasserstein distance-based intrinsic reward) when computing the reward-to-go $\hat{R}$ for updating each agent's critic network. The remaining components of MAPPO do not require modification. We conduct experiments on Pac-Men, SMAC, and SMACv2 to test the performance of WMAD+MAPPO. The results, presented in Table 1, demonstrate the superior performance of WMAD+MAPPO compared to the baselines.

## G  ENVIRONMENTAL DETAILS AND ADDITIONAL EXPERIMENTAL RESULTS

## G.1 ENVIRONMENTAL DETAILS AND EXPERIMENTAL RESULTS

In Pac-Men, four agents are initialized in the central room of a maze. Each agent is restricted to observing a 4×4 grid surrounding them. Randomly distributed dots are present in each edge room. The objective for the agent is to gather as many dots as possible in each edge room. We vary the lengths of paths to evaluate the exploration of environments. Specifically, path lengths for downward, leftward, rightward, and upward directions are set to 3, 6, 6, and 10, respectively. Only one path falls within the agent's observation scope. Dots in each room will respawn once all have been consumed by agents. Agents receive an environmental reward equal to the total number of dots consumed in each time step.

The SMAC benchmark includes many cooperative tasks based on Blizzard's real-time strategy game StarCraft II, designed to evaluate the efficacy of different Multi-Agent Reinforcement Learning (MARL) algorithms. Agent-level control in SMAC utilizes the Machine Learning APIs provided by StarCraft II and DeepMind's PySC2. Each task presents a combat scenario with two armies: one led by allied RL agents and the other by a non-learning game AI. The game ends when all units from any army perish or a predefined time limit is reached. The objective for allied agents is to maximize the game's win rate. To achieve this, agents must learn a sequence of actions to effectively collaborate with allies in vanquishing enemy forces. An illustrative example of such collaboration involves mastering kiting skills, where agents organize formations based on armor types to lure enemy units into pursuit while maintaining a safe distance to minimize damage. The SC2.4.10 version of StarCraft II is utilized, and performance comparison across different versions are not applicable. Experiments are conducted across six scenarios, including 3s5z, 2c_vs_64zg, 7sz, 6h_vs_8z, corridor, and 3s5z_vs_3s6z, spanning various difficulty levels.

SMAC is greatly limited by its lack of stochasticity. To remedy this, the newly released SMACv2 proposes modifications such as incorporating random team compositions and random start positions. These adjustments aim to inject more stochastic elements into the environment to effectively evaluate the exploration capabilities of MARL algorithms. We conduct experiments in three SMACv2 scenarios: terran_5_vs_5, protoss_5_vs_5, and zerg_5_vs_5. In SMACv2, each race in the game of StarCraft II employs three unit types, with units algorithmically assembled into teams. The probability of each unit type appearing in each episode remains fixed throughout training and testing phases. Allied agents have the same unit types as their adversaries. In each episode, allied agents are randomly deployed on the map using either a reflect or surround style.

We present the average returns of all algorithms in Pac-Men, SMAC, and SMACv2, along with their standard deviation over five random seeds, in Table 1. The results indicate the significant performance superiority of our method over baseline methods.

Table 1: Average returns of all algorithms in Pac-Men, SMAC, and SMACv2. $\pm$ denotes the standard deviation over five random seeds.

| Method | Pac-Men | SMAC | | | | | | SMACv2 | | |
|---|---|---|---|---|---|---|---|---|---|---|
| | | 3s5z | 2c_vs_64zg | 7sz | 6h_vs_8z | corridor | 3s5z_vs_3s6z | terran_5_vs_5 | protoss_5_vs_5 | zerg_5_vs_5 |
| QMIX | 0.21±0.04 | 0.72±0.13 | 0.85±0.08 | 0.17±0.02 | 0.23±0.03 | 0.57±0.07 | 0.36±0.12 | 0.68±0.03 | 0.53±0.05 | 0.41±0.04 |
| MAPPO | 0.49±0.03 | 0.81±0.05 | 0.83±0.04 | 0.52±0.06 | 0.53±0.03 | 0.62±0.05 | 0.57±0.08 | 0.52±0.04 | 0.47±0.03 | 0.37±0.03 |
| MAVEN | 0.32±0.06 | 0.51±0.21 | 0.72±0.06 | 0.00±0.00 | 0.42±0.04 | 0.36±0.08 | 0.18±0.15 | 0.58±0.04 | 0.31±0.05 | 0.29±0.03 |
| EOI | 0.41±0.05 | 0.87±0.07 | 0.83±0.02 | 0.37±0.03 | 0.08±0.03 | 0.25±0.11 | 0.42±0.13 | 0.65±0.05 | 0.42±0.03 | 0.47±0.04 |
| QTRAN | 0.28±0.08 | 0.21±0.19 | 0.75±0.05 | 0.00±0.00 | 0.02±0.02 | 0.08±0.07 | 0.02±0.01 | 0.42±0.02 | 0.40±0.04 | 0.25±0.02 |
| SCDS | 0.37±0.05 | 0.76±0.07 | 0.57±0.09 | 0.21±0.03 | 0.03±0.01 | 0.56±0.06 | 0.00±0.00 | 0.52±0.03 | 0.47±0.05 | 0.38±0.04 |
| PMIC | 0.34±0.03 | 0.82±0.03 | 0.79±0.05 | 0.58±0.02 | 0.51±0.05 | 0.37±0.03 | 0.18±0.06 | 0.47±0.03 | 0.36±0.02 | 0.42±0.02 |
| LIPO | 0.43±0.02 | 0.71±0.03 | 0.76±0.02 | 0.39±0.04 | 0.36±0.06 | 0.27±0.03 | 0.21±0.03 | 0.43±0.02 | 0.46±0.03 | 0.37±0.03 |
| FoX | 0.39±0.03 | 0.74±0.02 | 0.64±0.05 | 0.56±0.03 | 0.45±0.05 | 0.52±0.04 | 0.43±0.04 | 0.54±0.03 | 0.56±0.02 | 0.49±0.02 |
| **WMAD+QMIX** | **0.87±0.03** | **0.95±0.03** | **0.96±0.02** | 0.87±0.04 | **0.83±0.03** | 0.85±0.04 | 0.82±0.03 | 0.85±0.03 | **0.90±0.02** | 0.84±0.03 |
| **WMAD+MAPPO** | 0.82±0.02 | 0.93±0.02 | 0.89±0.05 | **0.94±0.03** | 0.79±0.04 | **0.87±0.05** | **0.89±0.04** | **0.89±0.03** | 0.82±0.02 | **0.91±0.04** |

## G.2 ADDITIONAL RESULTS

**Homogeneous behaviors** Agents may sometimes desire to behave in the same way. For instance, allied agents in the scenarios of SMAC might take the same action to fire at the same enemy in order to rapidly defeat it. In this section, to demonstrate the effectiveness of our method in learning such behaviors, we evaluate our method in four homogeneous scenarios of SMAC that require the trick of

Table 2: Performance of our method and QMIX in homogeneous scenarios.

| Method | 8m | 5m_vs_6m | 8m_vs_9m | 10m_vs_11m |
|---|---|---|---|---|
| WMAD+QMIX | 0.94±0.02 | 0.95±0.03 | 0.93±0.04 | 0.91± 0.03 |
| QMIX | 0.87±0.03 | 0.65±0.04 | 0.58±0.05 | 0.43±0.04 |

Table 3: Performance of our method and QMIX in scenarios of SMACv2 with different number of agents

| Method | terran_5_vs_5 | terran_10_vs_10 | terran_15_vs_15 | terran_20_vs_20 |
|---|---|---|---|---|
| WMAD+QMIX | 0.85±0.03 | 0.86 ±0.02 | 0.83 ±0.04 | 0.81 ±0.05 |
| QMIX | 0.68±0.03 | 0.39±0.04 | 0.24 ±0.06 | 0.11±0.05 |

focus fire. The results are shown in Table 2. Our method outperforms QMIX across all scenarios, demonstrating that our method would not prevent the homogeneous behaviors if they can lead to more environmental rewards. In contrast, our method encourages sufficient exploration to search for such optimal cooperative behaviors.

**Scalability** The scalability of the MARL algorithms refers to their ability to effectively handle the growing number of agents in the environment. The action space grows exponentially with the number of agents, highlighting the urgent need for exploration. In this section, we evaluate the scalability of our method in four scenarios of SMACv2 with an increasing number of agents: terran_5_vs_5, terran_10_vs_10, terran_15_vs_15, and terran_20_vs_20. We present the results in Table 3. Our method maintains its outperformance over QMIX across all scenarios. QMIX suffers from poor scalability due to limited exploration, while our method scales well with an increasing number of agents, demonstrating that our method can lead to sufficient exploration of action space by enlarging the Wasserstein distance between trajectory distributions of different agents in the latent representation space.

## H COMPARISON WITH $\epsilon$-GREEDY

The $\epsilon$-greedy method is a commonly used exploration strategy in many RL algorithms. Typically, increasing the value of $\epsilon$ enhances exploration. In this section, we compare our Wasserstein distance-based method with $\epsilon$-greedy to highlight its effectiveness in promoting exploration within MARL. For this comparison, we set the $\epsilon$ values to 0.05, 0.075, and 0.1 for QMIX, and evaluate these settings in the challenging scenarios including corridor, 3s5z_vs_3s6z, terran_5_vs_5, and protoss_5_vs_5. The results, presented in Table 4, show that our entropy maximization method is more effective in fostering exploration compared to simply increasing $\epsilon$. Notably, increasing $\epsilon$ values does not lead to significant performance gains. In multi-agent settings, higher $\epsilon$ values primarily increase randomness in an individual agent's action selection without enhancing diversity or coordination among agents, as they fail to consider the trajectories of other agents, resulting in inefficient exploration.

## I TRAINING DETAILS AND HYPERPARAMETERS

In this section, we provide the training details and hyperparameters adopted in our experiments. To implement CPC, we use a two-layer MLP with a hidden size of 64 for the encoder $g_{\theta_e}$ followed by the batch normalization and a GRU unit for the autoregressive model $g_{\theta_g}$. We adopt a dual vector with a dimension $m$ of 64 to parameterize the dual function. To integrate our method with QMIX, the intrinsic agent utility network is implemented with a two-layer MLP with a hidden size of 64. We keep other components the same as in QMIX.

Table 4: Comparison of performance between our method and QMIX using various $\epsilon$ values

| Method | corridor | 3s5z_vs_3s6z | terran_5_vs_5 | protoss_5_vs_5 |
|---|---|---|---|---|
| $\epsilon$ = 0.05 (QMIX) | 0.57 ±0.07 | 0.36 ±0.12 | 0.68 ±0.03 | 0.53 ±0.05 |
| $\epsilon$ = 0.075 (QMIX) | 0.61 ±0.04 | 0.39 ±0.11 | 0.72 ±0.04 | 0.62 ±0.07 |
| $\epsilon$ = 0.1 (QMIX) | 0.63 ±0.06 | 0.44 ± 0.15 | 0.74 ±0.03 | 0.69 ±0.06 |
| Wasserstein distance (our method) | 0.85 ± 0.04 | 0.82 ± 0.03 | 0.85 ±0.03 | 0.90 ±0.02 |

The policy networks of all agents are implemented with Deep Recurrent Q-Networks. At each time step, an agent's policy network processes a local observation as input, which is then forwarded through a fully-connected hidden layer, followed by a GRU unit, and ultimately a fully-connected layer generating $U$ outputs, where $U$ is the number of actions. Furthermore, all agents' policies share the same policy network parameters to accelerate training. We set the evaluation interval to 10K steps followed by 32 test episodes. We run all methods for 5 million steps in all tested tasks. We employ hard updates to update target networks every 200 episodes in SMAC and SMACv2. In Pac-Men, we utilize soft updates for updating target networks with a momentum of 0.01. The common hyperparameters are consistent across various methods for each multi-agent task. Detailed hyperparameters are provided in Table 5. The replay buffer size is set to 5K. We implement our method using NumPy and PyTorch. All experiments are performed on a NVIDIA GeForce RTX 4090 GPU.

Table 5: Hyperparameters

| | Pac-Men | SMAC | SMACv2 |
|---|---|---|---|
| hidden dimension | 64 | 128 | |
| learning rate | 0.0003 | 0.005 | |
| optimizer | | Adam | |
| target update | 0.01(soft) | 200(hard) | |
| batch size | 32 | 64 | |
| $\beta$ | 0.03 | 0.05 | |
| $\alpha$ for WMAD+QMIX | 0.01 | 0.005 for 3s5z, 2c_vs_64zg, 8m, 5m_vs_6m, 8m_vs_9m, and 10m_vs_11m, 0.05 for 7sz, 6h_vs_8z, corridor, and 3s5z_vs_3s6z | 0.03 |
| $\alpha$ for WMAD+MAPPO | 0.01 | 0.005 for 3s5z, 2c_vs_64zg, 8m, 5m_vs_6m, 8m_vs_9m, and 10m_vs_11m, 0.03 for 7sz, 6h_vs_8z, corridor, and 3s5z_vs_3s6z | 0.03 |
| epsilon anneal time | 200,000 | 200,000 for 3s5z, 2c_vs_64zg, 8m, 5m_vs_6m, 8m_vs_9m, and 10m_vs_11m, 500,000 for 7sz, 6h_vs_8z, corridor, and 3s5z_vs_3s6z | 500,000 |

## J    VISUALIZATIONS

Challenging tasks typically necessitate complex cooperative behaviors requring agents to learn diverse policies. We next present some visualization examples of diverse policies learned by our method in the super hard scenarios (6h_vs_8z, corridor, and 3s5z_vs_3s6z) in Figure 7. In the 6h_vs_8z scenario, one agent first leaves the team, causing most enemies to follow the lone agent's movements. The agent continues moving away to draw the enemies' fire and cover other agents. Other agents then quickly surround the few remaining enemies. Through learning such cooperative tactics, agents successfully scatter the enemies' powerful attacks. If all agents behave similarly and directly move towards enemies, they would be killed by enemies immediately. Similar tactics can also be observed in the other two scenarios, demonstrating the effectiveness of our method in encouraging multi-agent diversity.

We also present the visitation heatmaps of mutual information-based methods and our method in the protoss_5_vs_5 and the zerg_5_vs_5 scenarios in Figures 8 and 9, respectively. The visitation heatmaps reveal that our proposed WMAD leads to more sufficient exploration compared to the baselines. We believe this is because the mutual information objective does not provide effective incentives for exploring the environment. As a result, the agents trained by mutual information-based methods are slow to search for randomly appearing enemies on the map. In contrast, our method enables sufficient exploration by enlarging the Wasserstein distance.

## K    EVALUATIONS OF DIFFERENT KERNEL FUNCTIONS

We use the Gaussian kernel by default in our paper. We may also use a linear kernel to parameterize dual functions. To evaluate the effectiveness of using the linear kernel for dual functions, we design a linear kernel variant and test it in the super hard scenarios of SMAC. The results are shown in Table 6 .We note that using the linear kernel to parameterize dual functions leads to significant performance decline. We suspect this is because the dual function may not be linear functions. Using the linear kernel constraints the representation ability of the dual function.

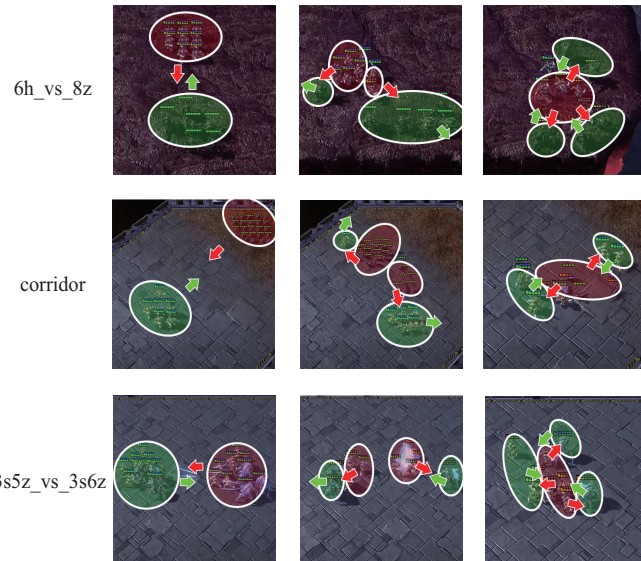

Figure 7: Visualization examples of diverse policies emerging in 6h_vs_8z (top), corridor (medium), and 3s5z_vs_3s6z (bottom) from initial (left) to final (right). Green and red shadows represent agents and enemies, respectively. Green and red arrows represent the moving directions of agents and enemies, respectively.

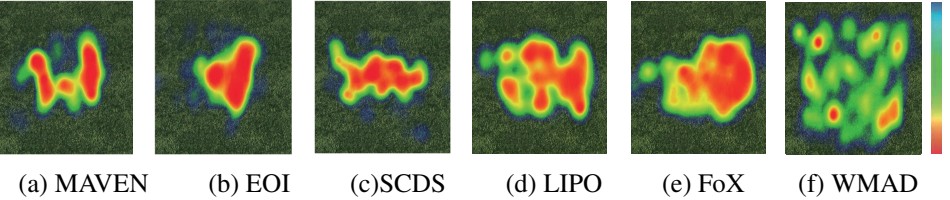

(a) MAVEN      (b) EOI      (c)SCDS      (d) LIPO      (e) FoX      (f) WMAD

Figure 8: Visitation heatmaps of different algorithms in the protoss_5_vs_5 scenario.

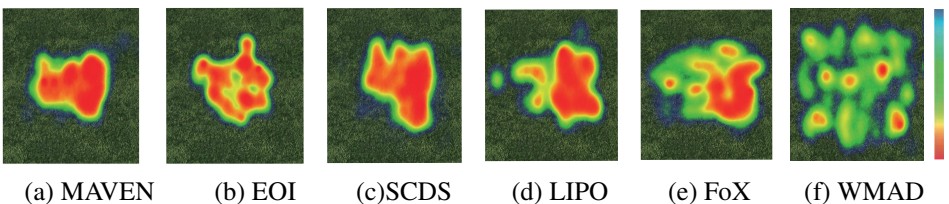

(a) MAVEN      (b) EOI      (c)SCDS      (d) LIPO      (e) FoX      (f) WMAD

Figure 9: Visitation heatmaps of different algorithms in the zerg_5_vs_5 scenario.

## L EVALUATIONS OF DIFFERENT VALUES FOR THE WEIGHT OF THE INTRINSIC REWARD $\alpha$

The values for the weight of the intrinsic reward $\alpha$ in different scenarios are listed in Table 5 in our paper. To investigate the effect of different weights of the intrinsic reward, we evaluate different weight values in the easy scenario 3s5z and the super hard scenario corridor. The results are shown in Table 7. The results demonstrate that our method is not very sensitive to the values of the weight. Sub-optimal weights do not result in a significant performance drop even in the super hard scenario.

Table 6: Performance comparisons of WMAD with different kernel functions in the scenarios of SMAC

| Methods | 6h_vs_8z | corridor | 3s5z_vs_3s6z |
|---|---|---|---|
| WMAD (Linear Kernel) | $0.57\pm 0.07$ | $0.39 \pm 0.05$ | $0.32 \pm 0.03$ |
| WMAD (Ours) | $0.85 \pm 0.03$ | $0.90 \pm 0.03$ | $0.87 \pm 0.04$ |

Table 7: Performance comparisons of WMAD with different values for the weight of the intrinsic reward $\alpha$.

| Methods | 3s5z | | | corridor | | |
|---|---|---|---|---|---|---|
| | $\alpha = 0.02$ | $\alpha = 0.05$ | $\alpha = 0.1$ | $\alpha = 0.02$ | $\alpha = 0.05$ | $\alpha = 0.1$ |
| WMAD | $0.89 \pm 0.03$ | $0.91 \pm 0.02$ | $0.93 \pm 0.03$ | $0.82 \pm 0.07$ | $0.85 \pm 0.04$ | $0.81 \pm 0.05$ |

## M  EVALUATIONS OF DIFFERENT COST FUNCTIONS

In our paper, we mainly use the Wasserstein distance to encourage sufficient exploration and simply adopt the Euclidean distance as the cost function as in many prior works. We may also use cosine similarity as the cost function, which measures the direciton differences between data points. We test the cosine similarity in Pac-Men, where agents need to move to different directions. The resutls are shown in Table 8. We note that the Wasserstein distance based on the cosine similarity achieves higher rewards in Pac-Men. In our work, we do not specifically discuss different cost functions and use the default Euclidean distance because we want to be consistent with prior works using the Wasserstein distance to ensure a fair comparison.

Table 8: Performance comparisons of WMAD using different cost functions.

| Method | Pac-Men |
|---|---|
| WMAD (Cosine Similarity) | $94 \pm 0.05$ |
| WMAD (Euclidean Distance) | $87 \pm 0.03$ |

