# OpenReview forum: "Unsupervised Multi-Agent Diversity With Wasserstein Distance"
_ICLR.cc/2025/Conference — Submitted to ICLR 2025_

### Official Review · Reviewer_VJC3 · 2024-10-20

**Soundness:** 3
**Presentation:** 2
**Contribution:** 3
**Rating:** 5
**Confidence:** 3

**Summary:**

This paper proposes Wasserstein Multi-Agent Diversity (WMAD), a new method for promoting exploration in Multi-Agent Reinforcement Learning (MARL). Unlike mutual information-based approaches, WMAD maximizes the Wasserstein distance between agents' trajectory distributions to encourage diverse behaviors. The method leverages Contrastive Predictive Coding (CPC) to learn trajectory representations and introduces a nearest neighbor intrinsic reward based on the Wasserstein distance. WMAD achieves more diverse policies and better exploration, outperforming state-of-the-art methods in complex multi-agent tasks.

**Strengths:**

- The paper introduces a new approach by using the Wasserstein distance to promote agent diversity, addressing the limitations of mutual information-based methods in encouraging effective exploration.

- Using contrastive predictive coding for learning distinguishable trajectory representations enhances the ability to measure differences between agents’ behaviors.

- The method is evaluated across multiple challenging multi-agent environments (like Pac-Men, SMAC, and SMACv2), demonstrating consistent outperformance over baseline methods. The proposed approach is integrated with MARL algorithms like QMIX, showing its practical applicability and potential to improve real-world cooperative learning tasks.

- The paper is organized clearly and it is easy to follow its core idea.

**Weaknesses:**

- The Wasserstein distance relies on an appropriate cost function to measure trajectory differences, and the paper uses a simple Euclidean distance without exploring task-specific alternatives, which may limit the method’s adaptability.

- Although the paper employs kernel-based techniques to reduce costs, computing the Wasserstein distance for every pair of agents in large-scale multi-agent systems can still be computationally intensive, making the system not scalable.

- The paper does not thoroughly explore the sensitivity of the method to key parameters, such as the choice of kernel or the weighting of intrinsic rewards, which could affect generalizability.

**Questions:**

- Have you explored alternative cost functions tailored to specific tasks, and if so, how did they impact the results?

- Although you adopted a kernel-based method to reduce computational costs, what challenges did you face in scaling the method to larger multi-agent systems? Will this limit the proposed method's scalability?

- How sensitive is your method to the selection of hyperparameters, such as the kernel width or the coefficient for intrinsic rewards? Did you conduct any sensitivity analysis to understand their impact?

- Since the effectiveness of your method heavily relies on CPC for trajectory representation, how robust is the learned representation to noise or perturbations in agent observations?

---

> ### Author Response · Authors · 2024-11-24
>
> Thanks for your feedback. We respond to your concerns below:
>
> Weakness1: The ... adaptability.
>
> In our paper, we use the Wasserstein distance to encourage sufficient exploration and simply adopt the Euclidean distance as the cost function as in many prior works. We may also use cosine similarity as the cost function, which measures the direction differences between data points. We test the cosine similarity in Pac-Men, where agents require to move in different directions.  The results are shown below:
>
> |  Method   | Pac-Men  |
> |---------------------------------------------|-------------------|
> |WMAD (Cosine Similarity) | 94 $\pm$ 0.05 |
> |WMAD (Euclidean distance) | 87 $\pm$ 0.03 |
>
> We note that the Wasserstein distance based on the cosine similarity achieves higher rewards. In our work, we do not specifically discuss different cost functions and use the default Euclidean distance because we want to be consistent with prior works using the Wasserstein distance to ensure a fair comparison. We have added this analysis to our paper.
>
> Weakness 2: Although ... not scalable.
>
> Our method is scalable, as demonstrated in the last paragraph of Appendix G.2. We evaluate our method in scenarios of SMACv2 with an increasing number of agents. Our method achieves satisfactory performance and scales well across all scenarios. Moreover, our method would not cost high computational resources. We provide comparisons of training time (5 million steps in the corridor scenario of SMAC) of our method against baselines in the table below:
>
> |  Method   | Training time  |
> |---------------------------------------------|-------------------|
> |QMIX | 7h 17m 30s |
> |MAVEN | 7h 23m 16s |
> |QTRAN | 8h 10m 28s |
> |EOI | 7h 46m 35s |
> |SCDS | 8h 23m 14s |
> |PMIC | 8h 59m 52s |
> |FOX | 7h 9m 52s |
> |LIPO  | 9h 10m 9s |
> |WMAD (Ours)  | 7h 35m 47s |
>
> Our method consumes relatively less training time compared to the baseline methods.
>
> Weakness 3: The ... generalizability.
>
> We use the Gaussian kernel by default in our paper. We may also use a linear kernel to parameterize dual functions. To evaluate the effectiveness of using the linear kernel for dual functions, we design a linear kernel variant and test it in the super hard scenarios of SMAC. The results are shown below:
>
> |  Method   | 6h\_vs\_8z  |corridor  |3s5z\_vs\_3s6z  |
> |---------------------------------------------|-------------------|------------------|------------------|
> |WMAD (Linear Kernel) | 0.57$\pm$ 0.07 |0.39 $\pm$ 0.05|0.32 $\pm$ 0.03|
> |WMAD (Ours) | 0.85 $\pm$ 0.03 |0.90 $\pm$ 0.03 |0.87 $\pm$ 0.04 |
>
> We note that using the linear kernel to parameterize dual functions leads to a significant performance drop. We suspect this is because the dual function may not be a linear function. Using the linear kernel constraints the representational ability of the dual function.
>
> The values for the weight of the intrinsic reward $\alpha$ in different scenarios are listed in Table 5 in our paper. To investigate the effect of different weights of the intrinsic reward, we test different weight values in the easy scenario 3s5z and the super hard scenario corridor. The results are shown in the table below:
>
> | Method    | 3s5z ($\alpha$=0.02)     | 3s5z ($\alpha$=0.05)     | 3s5z ($\alpha$=0.1)     | corridor ($\alpha$=0.02)  | corridor ($\alpha$=0.05)  | corridor ($\alpha$=0.1)   |
> |--------------------|------------------|------------------|-----------------|------------------|------------------|------------------|
> | WMAD   | 0.89 ± 0.03      | 0.91 ± 0.02      | 0.93 ± 0.03     | 0.82 ± 0.07      | 0.85 ± 0.04      | 0.81 ± 0.05      |
>
> The results demonstrate that our method is not very sensitive to the values of the weight. Sub-optimal weights do not result in a significant performance drop even in the super hard scenario. We have added these discussions in our paper.
>
> Q1:Have ... the results?
>
> See Weakness1;
>
> Q2: Although ... scalability?
>
> See Weakness2;
>
> Q3: How ... impact?
>
> See Weakness3;
>
> Q4:Since ... observations?
>
> We may add Gaussian noise to the observations and test the robustness of our representation learning method. We evaluate such a method in three super hard scenarios of SMAC. The results are shown below:
>
> |  Method   | 6h\_vs\_8z  |corridor  |3s5z\_vs\_3s6z  |
> |---------------------------------------------|-------------------|------------------|------------------|
> |WMAD (Gaussian noise) | 0.83 $\pm$ 0.05 |0.86 $\pm$ 0.08 |0.81 $\pm$ 0.09|
> |WMAD (Ours) | 0.85 $\pm$ 0.03 |0.90 $\pm$ 0.03 |0.87 $\pm$ 0.04 |
>
> We note that our method remains robust to noise in the observations. Our proposed representation learning method is based on the next-step prediction, which is more robust to direct representation learning from raw observations.

---

> > ### Author Response · Authors · 2024-12-01
> >
> > We greatly appreciate your suggestions and look forward to your feedback.

---

### Official Review · Reviewer_e3vP · 2024-10-26

**Soundness:** 3
**Presentation:** 3
**Contribution:** 2
**Rating:** 5
**Confidence:** 5

**Summary:**

This paper focuses on the issue of diversity in cooperative multi-agent reinforcement learning (MARL). As parameter sharing in MARL often leads to homogeneous behaviors and limited exploration, some previous methods promote identity-aware multi-agent diversity by mutual information (MI). The authors point out the drawbacks of MI and replace it with Wasserstein Distance. The Wasserstein Multi-Agent Diversity (WMAD) uses the Wasserstein distance between the trajectory distributions of different agents as an intrinsic reward to facilitate exploration. The authors conducted experiments on Pac-Men, SMAC and SMACv2 to test the proposed method.

**Strengths:**

1. The authors describe the framework and implementation of WMAD in detail, which makes the method easy to understand.
2. The motivation of this paper is very clear: First, promote multi-agent diversity for exploration; Second, improve previous mutual-information-based approaches.
3. The visualization of the visited area strongly demonstrates the effectiveness of WMAD in promoting diversity.

**Weaknesses:**

1. The novelty is relatively limited. As the authors mentioned, there are already many works about promoting diversity for enhanced exploration. WMAD follows them and replaces the metric of diversity with Wasserstein distance.
2. According to Figure 4(d), the diversity of agents' trajectories is improved significantly. However, how would WMAD perform in scenarios that require homogeneous behaviors (e.g., focus fire on the same enemy in SMAC)? I think the authors need to include results or discussion on WMAD's performance in such scenarios.
3. The experiment results in SMAC and SMACv2 are very significant. However, it is worth noting that the learning rate is set to 0.005 and the batch size is 128. The exploration rate is also tuned. These settings are proven to significantly improve the performance of QMIX in *pymarl2* [1]. Therefore, I wonder how the other baselines are implemented, and I'm concerned about the fairness of the experiments. Maybe the authors could clarify the implementation and hyperparameter settings of other baselines. Furthermore, It would be better to provide the results of WMAD under the hyperparameter settings in *pymarl* and *pymarl2*, respectively.

[1] Hu J, Jiang S, Harding S A, et al. Rethinking the implementation tricks and monotonicity constraint in cooperative multi-agent reinforcement learning[J]. arXiv preprint arXiv:2102.03479, 2021.

**Questions:**

1. I hope the authors can provide comparison results between WMAD and baselines in terms of the number of parameters and training costs.
2. Please see Weaknesses 3.

---

> ### Author Response · Authors · 2024-11-24
>
> Thank you for your careful review.
>
> Weakness 1: The ... distance.
>
> Promoting diversity for enhanced exploration is an emergent research direction. Our method solves the limitation of the mutual information-based method, which lacks diversity metric, and encourages sufficient exploration. Moreover, our contributions are different from prior works using the Wasserstein distance. Prior works overlook the problem of applying the Wasserstein distance in multi-agent settings, where the parameter-sharing policy network can lead to homogeneous policies, thereby undermining the proper functioning of the Wasserstein distance. The Wasserstein distance between any two agents' trajectory distributions approaches zero, i.e., $W(X,Y)\rightarrow 0$, where $X$ and $Y$ respectively represent the trajectory distributions of two agents. The ablation results show that simply using the Wasserstein distance without representation learning leads to a significant performance drop. To solve the problem, the main contribution of our method lies in representation learning with CPC as we discussed in the last paragraph of Section 1 in our paper. We consider a latent representation space to make the Wasserstein distance meaningful. We construct this representation space using the Contrastive Predictive Coding (CPC) method to learn distinguishable trajectory representations.
>
> Weakness 2: According ... scenarios.
>
> We analyze the performance of our method in scenarios requiring homogeneous behaviors in the last paragraph of Section 5.2: "Moreover, it is notable that our method also achieves satisfactory performance in the easy 3s5z scenario where agents sometimes need to behave in the same way to master the trick of 'focus fire', demonstrating that our method would not prevent the homogeneous behaviors that can lead to more environmental rewards. More experimental results related to such homogeneous behaviors can be found in Appendix G.2. These results reveal that our method efficiently balances exploration and exploitation, resulting in the learning of optimal cooperative policies."
>
> Weakness 3: The ... respectively.
>
> In Appendix I, we detail the hyperparameters and network structures used in our experiments. To ensure a fair comparison, we use consistent hyperparameters and the same network structure for all baselines. We implement our method based on PyMARL2. To compare the performance of our method under different settings, we test our method under the hyperparameter settings in PyMARL in three super hard scenarios of SMAC. The results are shown below:
>
>
> |  Method   | 6h\_vs\_8z  |corridor  |3s5z\_vs\_3s6z  |
> |---------------------------------------------|-------------------|------------------|------------------|
> |WMAD (PyMARL) | 0.79 $\pm$ 0.07 |0.93 $\pm$ 0.05 |0.83 $\pm$ 0.08|
> |WMAD (PyMARL2) | 0.85 $\pm$ 0.03 |0.90 $\pm$ 0.03 |0.87 $\pm$ 0.04 |
>
> We note that under the hyperparameter settings in PyMARL, our method achieves similar performance to that implemented with PyMARL2. This phenomenon demonstrates that our method is not overly sensitive to the hyperparameter values because of throughout exploration.
>
>
> Q1: I hope ... training costs.
>
> Our WMAD and the baseline methods including MAVEN, QTRAN, EOI, SCDS, PMIC, and FoX are based on the framework of QMIX. Additionally, MAVEN introduces a GRU unit and a discriminator that consists of a two-layer MLP. EOI learns an additional discriminator that consists of a two-layer MLP. Similar to EOI, SCDS also needs to learn a trajectory discriminator. PMIC introduces a Dual Mutual Information Estimator that includes a state encoder, an action encoder, and an action prediction network. LIPO is based on MAPPO and additionally introduces a variational trajectory discriminator for each agent to maximize the mutual information objective. Our method learns a trajectory encoder that consists of a two-layer MLP with a hidden size of 64 for the encoder and a GRU unit for the autoregressive model. We also adopt a dual vector with a dimension $m$ of 64 to parameterize the dual function. Note that our method has a comparable number of parameters to other methods but achieves significant performance improvement over baseline methods. Since all methods are evaluated under the same computational platform, we compare the training costs of various methods based on the training time (5 million steps in the corridor scenario of SMAC) that are shown below:
>
> |  Method   | Training time  |
> |---------------------------------------------|-------------------|
> |QMIX | 7h 17m 30s |
> |MAVEN | 7h 23m 16s |
> |QTRAN | 8h 10m 28s |
> |EOI | 7h 46m 35s |
> |SCDS | 8h 23m 14s |
> |PMIC | 8h 59m 52s |
> |FOX | 7h 9m 52s |
> |LIPO  | 9h 10m 9s |
> |WMAD (Ours)  | 7h 35m 47s |
>
> We note that our method requires relatively less training time compared to the baseline methods.
>
>
> We hope that the responses provided above have addressed your concerns. We would be grateful for your feedback.

---

> > ### Comment · Reviewer_e3vP · 2024-11-24
> >
> > I've read other reviewers' comments and the rebuttals, and I really appreciate the authors for their efforts. The experiment results of WMAD are convincing, and it seems the reviewers agree that the main issue of WMAD is about the novelty. Replacing the MI-based metric with another metric is not a significant innovation, and the application of CPC in reinforcement learning has already been discussed in its original paper. WMAD combines these techniques together like building blocks but lacks new theoretical findings. Therefore, I have to keep my score.

---

> > > ### Author Response · Authors · 2024-11-25
> > >
> > > Thank you for your response. Trajectory representation learning in our method is necessary. It enlarges the distance between trajectories in a latent space, enabling the proper functioning of the Wasserstein distance. We propose a novel next-step prediction method based on CPC to learn distinguishable trajectory representations. As a result, by using our method, we do not need to use per-agent policy networks to introduce heterogeneous behaviors, unlike previous works such as CDS [1] and DiCo [2], which significantly reduces the number of parameters. This idea is novel and has never been proposed in previous works.
> > >
> > > [1] Li, Chenghao, et al. "Celebrating diversity in shared multi-agent reinforcement learning." Advances in Neural Information Processing Systems 34 (2021): 3991-4002.
> > >
> > > [2] Bettini M, Kortvelesy, et al. “Controlling Behavioral Diversity in Multi-Agent Reinforcement Learning” International Conference on Machine Learning (2024)

---

### Official Review · Reviewer_C1n6 · 2024-10-30

**Soundness:** 3
**Presentation:** 3
**Contribution:** 2
**Rating:** 6
**Confidence:** 4

**Summary:**

In order to promote exploration in multi-agent reinforcement learning, the authors propose a method WMAD to maximize the difference in agents’ trajectories. The difference in trajectories is represented by Wasserstein distance, which is calculated with latent variables. Extensive experiments are conducted to show the superiority of WMAD in tasks including Pac-Men and SMAC.

**Strengths:**

1. The authors propose SMAC to promote exploration by using Wasserstein distance to evaluate the difference in trajectories of different agents, which is more reasonable than mutual information.
2. Experimental results show that the proposed algorithm WMAD is much better than baseline algorithms.

**Weaknesses:**

1. Although the use of Wasserstein distance is better than mutual information, it seems that the idea of using Wasserstein distance to enhance the difference in trajectories of agents has been proposed, such as “Controlling Behavioral Diversity in Multi-Agent Reinforcement Learning”.
2. The authors claim that the proposed algorithm WMAD achieve SOTA with better exploration, while the baseline algorithms in experiments are not specifically designed for exploration. Baselines are fundamental MARL algorithms and mutual information-based exploration algorithms. Other kinds of exploration methods are missing, such as “Episodic multi-agent reinforcement learning with curiosity-driven exploration”.
3. It seems that the results of baselines are much worse than those in original papers, such as MAVEN in 6h_vs_8z (super hard) and corridor (super hard).

**Questions:**

See the weaknesses. Look forward to more explanation.

---

> ### Author Response · Authors · 2024-11-24
>
> Thank you for the in-depth review. Here are the responses to your questions and concerns:
>
> Q1: Although ... Learning”.
>
> Our contributions are different from prior works that use the Wasserstein distance. Prior works overlooked the problem of applying the Wasserstein distance to multi-agent settings, where the parameter-sharing policy network may lead to homogeneous policies. Such homogeneous policies disable the proper functioning of the Wasserstein distance. The Wasserstein distance between any two agents' trajectory distributions approaches zero, i.e., $W(X,Y)\rightarrow 0$, where $X$ and $Y$ respectively represent the trajectory distributions of two agents. The ablation results show that simply using the Wasserstein distance without representation learning leads to a significant performance decline. To solve the problem, the main contribution of our method lies in representation learning with CPC as we discussed in the last paragraph of Section 1 in our paper. We consider a latent representation space to make the Wasserstein distance meaningful. We construct this representation space using the Contrastive Predictive Coding (CPC) method to learn distinguishable trajectory representations.
>
> Moreover, prior works do not consider the high computational cost caused by calculating the Wasserstein distance. We propose a novel kernel method to calculate the Wasserstein distance. Our method only needs to optimize two dual vectors, which significantly reduces the computational cost.
>
>
> Q2: The ... exploration”.
>
> We compare our method with EMC proposed in "Episodic multi-agent reinforcement learning with curiosity-driven exploration" in three super hard scenarios of SMAC. The results are shown below:
>
> |  Method   | 6h\_vs\_8z  |corridor  |3s5z\_vs\_3s6z  |
> |---------------------------------------------|-------------------|------------------|------------------|
> |EMC | 0.37 $\pm$ 0.05 |0.76 $\pm$ 0.08 |0.73 $\pm$ 0.04|
> |WMAD (Ours) | 0.85 $\pm$ 0.03 |0.90 $\pm$ 0.03 |0.87 $\pm$ 0.04 |
>
> Our method outperforms EMC, demonstrating the effectiveness of our Wasserstein distance-based exploration.
>
> Q3: It seems ... hard).
>
> The performance differences of baseline methods stem from several reasons: first, we use consistent hyperparameters and the same network structures for all baselines to ensure a fair comparison. The hyperparameters and network structures may differ from those used in the original papers; second, performance comparisons across different SMAC versions are not applicable. The settings of scenarios across different versions can be different.
>
>
> We hope the responses provided above have resolved your concerns. Your feedback would be greatly appreciated.

---

> > ### Comment · Reviewer_C1n6 · 2024-11-26
> >
> > Considering the efforts the authors have made in their response, I have decided to increase my score. However, through the author's response, the novelty is still somewhat incremental from my perspective, like other reviewers, especially with the comparison with other MARL algorithms including Wasserstein distance. Besides, I am uncertain about the effectiveness and stability of the performance gains achieved by the method proposed by the authors.

---

> > > ### Author Response · Authors · 2024-11-27
> > >
> > > Our method adopts a trajectory representation learning technique using novel next-step prediction to solve an emergency problem, where the homogeneous trajectories may not enable the proper functioning of the Wasserstein distance. Instead, we enlarge the Wasserstein distance between trajectories in a latent space. As a result, by using our method, we do not need to use per-agent policy networks to introduce heterogeneous behaviors, unlike previous works such as CDS [1] and DiCo [2], which significantly reduce the number of parameters. This idea is novel and has never been proposed in previous works.
> > >
> > > Our statically reliable results, which are average returns of all algorithms in Pac-Men, SMAC, and SMACv2 along with the standard deviation over five random seeds, demonstrate the effectiveness and stability of the performance gains achieved by our method.
> > >
> > > [1] Li, Chenghao, et al. "Celebrating diversity in shared multi-agent reinforcement learning." Advances in Neural Information Processing Systems 34 (2021): 3991-4002.
> > >
> > > [2] Bettini M, Kortvelesy, et al. “Controlling Behavioral Diversity in Multi-Agent Reinforcement Learning” International Conference on Machine Learning (2024)

---

> > > ### Author Response · Authors · 2024-12-01
> > >
> > > We hope the responses above have addressed your concerns. We would appreciate receiving your feedback.

---

### Official Review · Reviewer_z99x · 2024-11-04

**Soundness:** 2
**Presentation:** 4
**Contribution:** 2
**Rating:** 5
**Confidence:** 5

**Summary:**

This paper proposes a novel approach to multi-agent policy diversity within the MARL domain. Firstly, the paper provides a detailed analysis of the shortcomings of current diversity methods based on mutual information. Subsequently, it leverages a CPC-based next-step prediction method to facilitate the learning of distinguishable representations of agent trajectories. Furthermore, it introduces a method for rapidly calculating the Wasserstein distance in multi-agent systems, which is integrated into practical MARL algorithms in the form of intrinsic rewards. Finally, the effectiveness of the proposed method is validated on the Pac-Men, SMAC/SMACv2 environments.

**Strengths:**

1. The paper is clearly articulated and well-structured, with the discussion on MI-based methods being particularly enlightening.
2. The discussion in the experimental section is comprehensive, with a thorough design of ablation studies.

**Weaknesses:**

Although the paper is well-written, I have major concerns regarding the novelty of the paper.
1. The first is the introduction of Wasserstein Distance (WD) to quantify the policy diversity (as represented by trajectories) among agents, where there has already been related work in the MARL domain, which may not represent a significant innovation. For example, work [1] introduces the concept of system neural Diversity based on WD, and work [2] proposes a policy distance concept also based on WD by learning representations of policy distributions.
2. The second concern is about translating the diversity's WD into intrinsic rewards to encourage diversity. In fact, methods purely encouraging diversity are not limited to intrinsic rewards or objective functions but also include controlling the network structure. Work [3] has even gone beyond merely encouraging diversity to being able to control the diversity of a multi-agent system to a specific value. Therefore, this work might not be novel enough to match the ICLR community.
3. Correspondingly, there are concerns regarding the selection of baselines. Since this is a method encouraging multi-agent diversity, why has it only been compared with MI-based methods? Baselines should include MARL diversity-related but MI-unrelated methods. For example, RODE [4], ADMN[5], and the previously mentioned methods?

*I hope the authors can understand my concerns and address them together with the following questions.*


[1] Bettini M, Shankar A, et al. “System neural diversity: measuring behavioral heterogeneity in multi-agent learning”[J]. arXiv preprint arXiv:2305.02128, 2023.

[2] Hu T, Pu Z, Ai X, et al. “Measuring Policy Distance for Multi-Agent Reinforcement Learning” International Conference on Autonomous Agents and Multiagent Systems (2024)

[3] Bettini M, Kortvelesy, et al. “Controlling Behavioral Diversity in Multi-Agent Reinforcement Learning” International Conference on Machine
Learning (2024)

[4] T. Wang, T. Gupta, A. Mahajan, et al. “RODE: Learning Roles to Decompose Multi-Agent Tasks” International Conference on Learning Representations （2021）

[5]Yu Y, Yin Q, Zhang J, et al. “ADMN: Agent-Driven Modular Network for Dynamic Parameter Sharing in Cooperative Multi-Agent Reinforcement Learning”
International Joint Conference on Artificial Intelligence （2024）

**Questions:**

Beyond the major concerns I have listed, there are the following questions:
1. Can this method be applied to agents in continuous action spaces, and to multi-agents with different action spaces?
2. Regarding the discussion in lines 171-173 of the paper, can the authors provide an example to illustrate this point, and why wouldn't the WD directly become zero?
3. Compared to vanilla methods like QMIX and QTRAN, WMAD will undoubtedly introduce additional computational overhead. If WMAD appears to require fewer timesteps to achieve comparable performance levels, but demands more CPU/GPU computation time （or real time）, could this impact its practical use? Have there been any experiments conducted to assess the extent of this additional computational load?

4. WMAD chooses the Euclidean distance as the cost function to compute the Wasserstein distance. I am curious about the results if the Euclidean distance were used directly as the intrinsic reward.

---

> ### Author Response · Authors · 2024-11-24
>
> We sincerely appreciate your thorough review and valuable feedback on our manuscript. We answer your questions below:
>
> Weakness 1: The ... policy distributions.
>
> Our contributions are different from prior works that use the Wasserstein distance. Previous works overlook the problem of applying the Wasserstein distance in multi-agent settings, where the parameter-sharing policy network may result in homogeneous policies, hindering the proper functioning of the Wasserstein distance. The Wasserstein distance between any two agents' trajectory distributions approaches zero. The ablation results show that simply using the Wasserstein distance without representation learning leads to a significant performance decline. To solve the problem, the main contribution of our method lies in representation learning with CPC as we discussed in the last paragraph of Section 1 in our paper. We consider a latent representation space to make the Wasserstein distance meaningful.
>
> Moreover, prior works do not consider the high computational cost caused by calculating the Wasserstein distance. This is very important in multi-agent settings. High computational cost may lead to poor scalability. We propose a novel kernel method to calculate the Wasserstein distance. Our method only needs to optimize two dual vectors, which significantly reduces the computational cost.
>
>
> Weakness 2: The second ... ICLR community.
>
> We think extensive exploration is better than limited exploration. Control the diversity may lead to limited exploration and does not necessarily lead to better performance. Moreover, our method can achieve satisfactory performance in scenarios requiring homogenous behaviors, as demonstrated by Appendix G.2, indicating that our method efficiently balances exploration and exploitation.
>
> Weakness3: Correspondingly, ... methods?
>
> We compare our method with DiCo, RODE, and ADMN in three super hard scenarios of SMAC. For a fair comparison, we implement baseline methods with consistent hyperparameters and the same network structure.
>
> |  Method   | 6h\_vs\_8z  |corridor  |3s5z\_vs\_3s6z  |
> |---------------------------------------------|-------------------|------------------|------------------|
> | DiCo | 0.72 $\pm$ 0.04 |0.81 $\pm$ 0.03 |0.65 $\pm$ 0.09 |
> |RODE | 0.59 $\pm$ 0.08 |0.68 $\pm$ 0.08 |0.47 $\pm$ 0.11|
> |ADMN | 0.63 $\pm$ 0.05 |0.74 $\pm$ 0.06 |0.72 $\pm$ 0.08|
> |WMAD (Ours) | 0.85 $\pm$ 0.03 |0.90 $\pm$ 0.03 |0.87 $\pm$ 0.04 |
>
> The experimental results demonstrate the outperformance of our method compared to baseline methods.
>
> Q1: Can ... spaces?
>
> Our method can be used in multi-agent environments with continuous action spaces. We test our method in Cooperative Navigation, a multi-agent task with continuous action space, where agents learn to cooperatively cover all the landmarks while avoiding collisions.
>
> |  Method   | Average distance  |collisions |
> |---------------------------------------------|-------------------|------------------|
> |QMIX | 3.21 $\pm$ 0.15 |1.39 $\pm$ 0.27 |
> |WMAD (Ours) | 2.17 $\pm$ 0.11 |0.85 $\pm$ 0.18 |
>
> Compared to QMIX, our method leads to smaller average distances and fewer collisions.
>
> We evaluated our method in SMACv2 in our paper. SMACv2 is a benchmark with stochastic scenarios where agents have different action spaces.
>
> Q2: Regarding ... zero?
>
> We first consider an ideal condition. Due to policy network parameter sharing, the action distributions output by the policy network follow the same distribution $p$. As a result, $W(p, p) =0$. However, in practice, during the exploration phase, some techniques such as $\epsilon$-greedy have been used to improve the uncertainty in action selections. Thus, the distributions of different agents may not be identical. As a result, the Wasserstein distance approaches zero.
>
> Q3: Compared ... load?
>
> To reduce the computational cost, our method uses a kernel method, which only needs to learn two dual vectors. We compare the training time (5 million steps) of QMIX, WMAD w/ kernel method, and WMAD w/ two-layer neural network in the corridor scenario of SMAC under the same computation platform.
>
>
> |  Method   | Training time |
> |---------------------------------------------|-------------------|
> |QMIX | 7h 17m 30s |
> |WMAD w/ kernel method | 7h 35m 47s |
> |WMAD w/ two-layer neural network | 9h 29m 16s  |
>
> Our method does not cost much additional training time. However, using a two-layer neural network to parameterize the dual function leads to high computational cost.
>
> Q4: WMAD ... reward.
>
> The Euclidean distance directly measures the distance between data points, which may lead to high variance. We design a variant using the Euclidean distance as intrinsic rewards and test it in the super hard 3s5z\_vs\_3s6z scenario. This variant achieves an even lower win rate of 0.17 $\pm$ 0.09 compared to QMIX, which achieves a win rate of 0.36 $\pm$ 0.12.
>
> We hope to hear from you soon and thank you again for your review.

---

> > ### Comment · Reviewer_z99x · 2024-11-25
> >
> > Thank you for your response. Your reply has addressed some of my concerns. I particularly appreciate the addition of comparisons with methods such as Dico, RODE, and ADMN. However, I am not entirely satisfied with the rebuttal regarding W1. Additionally, could the authors provide some experimental details comparing with DiCo? It would be even better if the code (including the SMAC environment and the algorithm itself) could be shared via an anonymized link. Overall, I have decided to increase my score to 5（The highest I can give currently).

---

> > > ### Author Response · Authors · 2024-11-27
> > >
> > > Thank you for your response.
> > >
> > > We first clarify the differences between our method and previous works such as [1] and [2] to demonstrate our contributions.
> > >
> > > The authors in [1] propose controlling the heterogeneous policies via rescaling SND, which is measured by the Wasserstein distance. To produce heterogeneous policies, they introduce per-agent networks to learn diverse behaviors, which may lead to high computational cost and poor scalability. However, in our paper, we do not need additional per-agent networks, through contrastive representation learning, our method learns distinguishable trajectory representations that enable the Wasserstein distance to efficiently encourage multi-agent diversity. Moreover, we encourage extensive exploration. As we discussed in the response above, controling the diversity may not promote extensive exploration.
> > >
> > >
> > > The authors in [2] propose using the Wasserstein distance to measure the policy differences. Different from our contrastive representation learning, they use an encoder-decoder structure to learn latent representations of policies to standardize the action distributions of different agents. They ignore the problem that the Wasserstein distance with homogenous policies may not work properly.
> > >
> > > Moreover, as we discussed above, both works do not consider the high computational cost of the Wasserstein distance. In our method, we propose a novel kernel method to calculate the Wasserstein distance. Our method only needs to optimize two dual vectors, which significantly reduces the computational cost.
> > >
> > >
> > > The experiments on performance comparison between our method and DiCo follow the training details provided in Appendix H. We are happy to provide the code. However, we are not sure whether providing a link to our code violates the rule of Double-blind review. We need further confirmation from the Area Chair.
> > >
> > >
> > >
> > >
> > >
> > > [1] Bettini M, Shankar A, et al. “System neural diversity: measuring behavioral heterogeneity in multi-agent learning”[J]. arXiv preprint arXiv:2305.02128, 2023.
> > >
> > > [2] Hu T, Pu Z, Ai X, et al. “Measuring Policy Distance for Multi-Agent Reinforcement Learning” International Conference on Autonomous Agents and Multiagent Systems (2024)

---

### Meta-Review · Area_Chair_hDCh · 2024-12-22

**Metareview:**

This paper proposes Wasserstein Multi-Agent Diversity (WMAD), a new method for promoting exploration in Multi-Agent Reinforcement Learning (MARL). Unlike mutual information-based approaches, WMAD maximizes the Wasserstein distance between agents' trajectory distributions to encourage diverse behaviors. The method leverages Contrastive Predictive Coding (CPC) to learn trajectory representations and introduces a nearest neighbor intrinsic reward based on the Wasserstein distance. WMAD achieves more diverse policies and better exploration, outperforming state-of-the-art methods in complex multi-agent tasks.

The main concern shared among reviewers is the novelty, as WMAD just replaces the existing diversity-promoting methods with Wasserstein distance. The AC agrees and thus recommends rejection.

**Additional Comments On Reviewer Discussion:**

The main concern shared among reviewers is the novelty, as WMAD just replaces the existing diversity-promoting methods with Wasserstein distance. This concern was not fully addressed in the rebuttal.

---

### Decision · Program_Chairs · 2025-01-22

Reject